# An Observational Cross-Sectional Study of Gender and Disability as Determinants of Person-Centered Medicine in Botulinum Neurotoxin Treatment of Upper Motoneuron Syndrome

**DOI:** 10.3390/toxins14040246

**Published:** 2022-03-30

**Authors:** Cristina Maria Del Prete, Mattia Giuseppe Viva, Stefania De Trane, Fabrizio Brindisino, Giovanni Barassi, Alessandro Specchia, Angelo Di Iorio, Raffaello Pellegrino

**Affiliations:** 1Department of Physical Medicine and Rehabilitation, ASL, 73100 Lecce, Italy; cridelprete@tin.it; 2Department of Anatomical and Histological Sciences, Legal Medicine and Orthopaedics, Sapienza University of Rome, 00183 Rome, Italy; mattiagiuseppe.viva@uniroma1.it; 3Istituto di Ricovero e Cura a Carattere Scientifico (IRCCS), ICS Maugeri, 70124 Bari, Italy; stefania.detrane@icsmaugeri.it; 4Department of Medicine and Health Science “Vincenzo Tiberio”, University of Molise c/o Cardarelli Hospital, C/da Tappino, 86100 Campobasso, Italy; fabrizio.brindisino@unimol.it; 5Antalgic Mini-invasive and Rehab-Outpatients Unit, Department of Medicine and Science of Aging, University “G. d’Annunzio” Chieti-Pescara, 66100 Chieti, Italy; g.barassiunich.it@gmail.com (G.B.); raffaello.pellegrino@ucm.edu.mt (R.P.); 6Villa Beretta Rehabilitation Center, Valduce Hospital, Costa Masnaga, 23845 Lecco, Italy; alespecchiamd@gmail.com; 7Department of Scientific Research, Campus Ludes, Off-Campus Semmelweis University, 6912 Lugano–Pazzallo, Switzerland

**Keywords:** botulinum toxin, upper motor neuron syndrome, spasticity, sex-gender

## Abstract

The motor behaviour of patients with Upper Motor Neuron Syndrome (UMNS) is characterised by spasticity. The first-line treatment for this clinical condition is Botulinum neurotoxin A (BoNTA), but the number and key locations of muscles which need to be treated is not much discussed in the literature. Cross-sectional analysis of outpatient cohort with UMNS spasticity, who were potential candidates for BoNTA treatment, was performed. Between November 2020 and November 2021, all consecutive adult patients eligible for BoNTA treatment were enrolled. The inclusion criteria encompass UMNS spasticity (onset being ≥6 months), with disabling muscles hypertonia. Patients underwent a clinical evaluation, a comprehensive assessment with the Modified Ashworth Scale, with the Modified Rankin Scale, and a patients’ perception-centred questionnaire. In total, 68 participants were enrolled in the study, among them 40 (58.8%) were male; mean age 57.9 ± 15.1. In women, BoNTA was more frequently required for adductor group muscles, independently from potential confounders (OR = 7.03, 95%CI: 1.90–25.97). According to the pattern of disability, patients with hemiparesis more frequently need to be treated in the upper limb, whereas the diplegia/double-hemiparesis group needed to be treated more frequently at the adductor and crux muscles compared to their counterparts. UMNS spasticity in women could require more attention to be paid to the treatment of adductor muscle spasticity, potentially because the dysfunction of those muscles could influence sphincteric management, required for perineal hygiene and/or sexual life.

## 1. Introduction

The nerves in the central nervous system which carry the impulses for movement (sensorimotor networks and descending motor tracts) are known as upper motor neurons (UMN). Injury or lesions to UMNs are very common because of the vast areas covered by the motor neuron pathways.

Damage to the sensorimotor networks and descending motor tracts results in a neurological condition defined as Upper Motor Neuron Syndrome (UMNS) [1]. The motor behaviour of people with UMNS is characterised by a highly variable mixture of symptoms, ranging from lack of voluntary movement control, also called negative signs, combined with involuntary muscle contractions, called positive signs [2]. Spasticity is considered as a positive sign of UMNS and has been defined as a motor disorder characterised by a velocity-dependent increase in tonic stretch reflexes (muscle tone) with exaggerated tendon jerks, resulting from hyperexcitability of the stretch reflex [3,4]. Spasticity occurs commonly in a variety of diseases and conditions, including cerebral palsy, spinal cord injury or traumatic brain injury, Multiple Sclerosis (MS), and as a consequence of stroke, and according to different diseases, spasticity is estimated to affect between 17% to 53% of patients [1,5]. Spasticity is often accompanied by muscle discomfort and pain, and if untreated, can lead to contractures which cause mobility impairment, falls, and pressure ulcers, with a reduction in the perceived quality-of-life. One of the main goals for clinicians has always been to use available treatment strategies to improve muscle imbalance and joint alignment featured in the UMNS patterns, managing muscle overactivity to better cope with the negative consequences linked to spasticity. In other words, the principal objective of spasticity management is to reduce muscle overactivity in order to better control the consequences linked to that.

Botulinum neurotoxin A (BoNTA) is an effective treatment for many neurological disorders [6]. It has been largely studied and used as a treatment for focal spasticity [7]. In a meta-analysis investigating the efficacy and safety of BoNTA for upper limb spasticity following stroke and traumatic brain injury [8], BoNTA therapy improved muscle tone, reduced disability assessment scale, and increased patients’ global quality of life. In two other meta-analyses performed in patients with lower limb spasticity, the authors highlighted how BoNTA therapy produces long term reduction in muscle tone measured on the Fugl-Meyer scale score compared with a placebo control group [9,10]. Lastly, Esquenazi et al. reviewed evidence from the published literature for the treatment of upper and lower limb spasticity due to UMNS; in both cases, the evidence supported a high level recommendation for BoNTA in the treatments of UMNS spasticity [5]. The treatment of adult spastic paresis now tends to focus on the treatment of muscle overactivity rather than a comprehensive approach to care; however, person-centred care is increasingly adopted by healthcare systems in a shift of focus from “disease-oriented” towards “person-centred” medicine [11]. The current evidence supports the need to ensure that treatment interventions for spastic paresis should be centred, as far as reasonable, on the patient’s own priorities for treatment. Moreover, women have a higher risk compared to men in developing difficulties in performing basic activities of daily living following UMNS, and these sex differences increase with advancing age (the male–female health survival paradox) [12]. Lastly, a large body of evidence suggests the need for more focused studies assessing the sex-specific pharmacological responses in accordance with sex-gender-based medicine (SGSM) [13].

Gender Medicine represents a modern scientific research approach both to spread the knowledge already acquired on the sex-specific biological differences between males and females, and to promote the deepening of gender-specific indicators which can affect the health of the population.

In the past, sex and gender have been neglected; investigators have mainly studied only one sex [13]. Thus, physicians have had to extrapolate medical recommendations from research mainly done on men, slowing down the overall implementation of sex-gender difference detection in clinical research [13].

To the best of our knowledge, to date, there are no studies that have investigated sex differences in the treatment of UMNS spasticity with BoNTA. Therefore, the objective of this study is to evaluate whether there is a sex-gender-related difference between the number and location of muscles treated as per person needs, in a cohort of adult participants with UMNS related spasticity, attending the BoNTA outpatient clinic.

## 2. Results

A total of 68 participants were enrolled in the study, among those 40 (58.8%) were male; the mean age of the whole study group was 57.9 ± 15.1 years. Whilst the male group was slightly older (60.2 ± 16.3; vs. 54.6 ± 12.9) compared to female patients, the difference was not statistically significant (*p*-value = 0.13).

In Figure 1, disease prevalence is reported, according to sex. As could be expected, male patients were affected more frequently by stroke, whereas female reported more frequently MS (*p*-value for trend = 0.006). Hemiparesis was the most prevalent clinical pattern in 41 participants (60.3%). The frequency of reported hemiparesis in male was three time higher than in females (75.6% in men compared to 24.4% in women; OR = 6.20, 95%CI: 2.12–18.10).

According to the pattern of disability, the female group more frequently reported diplegia (6/9 66.7%) and double hemiparesis (12/18 66.7%). Globally, for both sexes, half of the patients enrolled in the study (*n* = 34/68, 50.0%) subjectively reported that both superior and inferior limbs could interfere with everyday physical function, whereas 34.4% (*n* = 22/68) of patients reported that only the lower limb limited their activities of daily living and 17.7% (*n*= 12/68) reported only the upper limbs as disability determinants.

No gender differences were found with regard to the reported upper limb disability (male 12.5%; female 10.7%), pain during movement (male 5%; female 3.6%), or resting pain (male 7.5%; female 14.2%). A similar figure was found in the recorded response with regard to the lower limb items.

According to MRS, no statistically significant differences (*p*-for-trend-value = 0.20) could be reported between sexes; 15 participants (22.1%) reported moderate disability, whereas 27 (39.7%) and 26 (38.2%) reported moderate/severe and severe disability, respectively (Figure 2).

In female sex, the BoNTA treatment for the adductor group muscles was more frequently required, which was independent from age, urine incontinence symptoms, menstruation cycle, and diagnosis (OR = 5.33, 95%CI: 1.42–20.09). Male sex required BoNTA treatment for the triceps, wrist flexors, and plantar flexors muscles more frequently; however, these differences were not found to be statistically significant (Table 1). Additionally, the mean number of treated muscles did not differ between sexes, (3.28 ± 1.39 and 3.39 ± 1.69 for male and female, respectively, adjusted for age *p*-value = 0.62).

With regards to the muscle groups injected, taking into account the pattern disability, (namely hemiparesis vs. diplegia/double hemiparesis), patients with hemiparesis more frequently needed to be treated in the upper limb (shoulder, forearm, and wrist), whereas the diplegia/double hemiparesis group need to be treated more frequently at the adductor and crux muscles level compared to their counterparts, independent of age and sex (Table 2). The difference in the mean number of muscles treated was not statistically significant between the two groups (Table 2).

The mean number of muscles treated also did not differ when considering the disability level: patients with severe disability (MRS = 5: mean treated muscles: 3.31 ± 1.44) compared to those with moderate/severe disability (MRS = 4, mean treated muscles: 3.44 ± 1.45; for the comparison between MRS5 and MRS4, *p*-value = 0.85) and moderate disability (MRS = 3, mean treated muscles: 3.01 ± 1.79; for the comparison between MRS5 and MRS3, *p*-value = 0.69).

With regards to the botulinum toxin, we treated 12 patients with Onabotulinum and 56 with Incobotulinum. The mean dose was 450 IU per subject for both toxins, ranging between 200 IU both for Onabotulinum and Incobotulinum to a maximum dose of 600 for Onabotulinum and 800 IU for Incobotulinum. No differences were detected between the use of the two drugs, and no side effects were reported.

## 3. Discussion

This study evaluated which muscle groups were required to be treated with BoNTA injection to reduce spasticity in an unselected cohort of patients affected by UMNS in a real-world therapeutic environment. In detail, females were four times as likely to require hip-adductor treatment. Whilst males were seven times more likely to require BoNTA treatment for the triceps, the low incidence in this muscle group overall meant that sex differences were not significant. Moreover, hemiparesis was associated with more frequently treatment for upper limb function, whereas diplegia/double hemiparesis required more clinical attention to the lower limbs. Lastly, the disability degree, according to the MRS score did not influence the mean number of treated muscles.

The physical appearance of individuals with physical disabilities influences body self-perceived thinking, and as a result, may lead to one’s negative or positive feelings and thoughts [14]. Additionally, gender, family status, and the severity level of the disability were found to be associated with self-concept, body image, and Quality of Life (QoL) [15,16].

Recently, BoNTA treatment for muscle spasticity was assessed as a determinant of QoL; in this study, the authors found an improvement in the QoL measured with a non-specific scale, whereas with another scale, which refers more directly to cornerstones of QoL, a significant improvement was not observed [17]. However, the BoNTA effects on QoL may be more specific and attributable to more complex processes, as also suggested by our results. As a matter of fact, our results demonstrated that women need BoNTA therapy more frequently for muscles involved in flexion and abduction of the hip, independently from level and pattern of disability as well as degree of spasticity. In other words, the location of muscles which required treatment was not only dependent on the clinical features of the disability, but was also sex-specific. Therefore, the functional requirements of women, in everyday life, could play a pivotal role in the pathway from spasticity to self-perceived disability. At least two factors should be accounted for in this contest: hygiene, and sexual life.

Safely managed drinking water, sanitation, and hygiene (WASH) supports general health, prevents disease, and enables participation in major life areas such as education and employment [18]. This becomes even more important in people who experience incontinence, and women and girls who menstruate may have further unmet requirements for water and adequate WASH facilities [19]. In our study, subsample of people with incontinence could not confound, but only mitigate the strength of the association between the muscles BoNTA-treated choice and the female sex. On the other hand, menstruation also did not interfere with the muscles BoNTA-treated choice, since this potentially confounder also did not change the strength of the association in multivariate analysis. Therefore, in our cohort, WASH was not clearly linked to the choice of muscles to be treated.

Sexual dysfunction correlates negatively with physical and psychosocial wellbeing in UMNS spasticity [20], and almost 75% of the UMNS-population reported a low satisfaction with sexual life [21]. Spasticity in hip adductors in both men and women may preclude usual sexual desire and sexual function [22]. Our study does not include a specific assessment of sexual life satisfaction; therefore, we could only speculate a role of sexual need in the choice of muscles to be treated in women. Our results could suggest that adductor spasticity treatment would be an unmet need for women affected by UMNS spasticity.

The results of this study revealed that pattern of paresis is another independent determinant of BoNTA muscles treatment choices. As could be expected, our data confirm that spasticity following hemiparesis commonly involves the muscles over the shoulder girdle [23]. Accordingly, the spasticity of such muscles frequently leads to painful/limited shoulder motions [24] requiring BoNTA treatment. Moreover, in hemiparetic patients, the nonimpaired lower limb could compensate for mobility disability, and in addition or alternatively, impaired lower limb function could be compensated for by greater proximal joint forces [25,26]; therefore, the patients’ need could be focalised in the search for autonomy in the upper function.

## 4. Limitation

Some limitations must be accounted discussing the results of this study. Obviously, the cross-sectional design of the study limits the assessment of cause-effect relationship; moreover, a selection bias (sampling bias) could be introduced due to the patients’ enrolment in an outpatient service.

Moreover, no ethical committee approval was requested, but in our observational study, we have only revised clinical data of routinary activities of an outpatient service.

The external validity of our results could be influenced by several factors; first of all, the lack of sample size calculation that could compromise the generalisability of our findings. Moreover, those data could reflect only a specific experience of patients of a specific area, and aimed to investigate the influence of a limited number of variables, limiting de facto the value of our results in a more widespread population. Lastly, the clustering of diplegia/double hemiparesis in the same group in the analytical process needs to be accounted for, as that could misrepresent/alter the associations found.

To best of our knowledge, this is the first paper to assess the needs of patients that could benefit from BoNTA treatment according to sex and disability, and this is a strength of our study.

## 5. Conclusions

In the treatment of female UMNS spasticity, more attention to hip flexor and adductor muscle spasticity could be advisable. Potentially, dysfunction of those muscles could influence sphincteric management required for perineal hygiene and/or sexual life. Larger and multicentre studies are needed to better characterise sex-specific patient needs.

## 6. Methods

A cross-sectional analysis of an outpatient cohort study of participants affected by UMNS spasticity, who were eligible candidates for BoNTA treatment, was performed. Patient recruitment took place at the Botulinum Toxin Outpatient Clinic of the Local Health Authority—Lecce, in the south of Italy. The study design was conceptualised at the Istituto di Ricovero e Cura a Carattere Scientifico -IRCCS, Bari, ICS Maugeri, Italy, and at the Antalgic Mini-invasive and Rehab-Outpatients Unit, Department of Medicine and Science of Aging; University “G. d’Annunzio” Chieti-Pescara, Italy.

Between November 2020 and November 2021, all consecutive participants over 18 years of age, affected by UMNS spasticity and who cyclically underwent physical and rehab treatment, who were potentially eligible for BoNTA treatment, were enrolled in the study. The study was developed following the Good Clinical Practice guidelines [27]. It was conducted in accordance with the ethical principles outlined in the Declaration of Helsinki, and with the procedures defined by the ISO 9001-2015 standards for “Research and experimentation”. Written informed consent was obtained at enrolment from participants who were willing and able; alternatively, informed consent was obtained from caregivers. The data collection and the procedures applied were part of the standard clinical routine; therefore, the normal ethics committee clearance was not required.

The inclusion criteria encompassed: UMNS spasticity due to ischemic or haemorrhagic stroke, MS, and outcomes of brain neoplasms or severe cranio-encephalic trauma, with disabling muscle hypertonia of grade 1+ to 3 on the Modified Ashworth Scale (MAS), and time from onset ≥6 months. Exclusion criteria were one or more of the following: presence of fixed contractures/or bone deformities of the limbs, MAS score ≥4, concomitant neuropathies, previous surgical interventions on the treated muscle groups, movement disorder, spinal cord injury, and medullary lesions. All patients who presented contraindications to injection treatment with BoNTA, such as drug allergies, ongoing skin infections at the infiltration site, and pregnancy, were also excluded from the study.

In the study period, 121 patients were consecutively evaluated, having been referred by general practitioners to the outpatient service; of those, 53 were excluded from the study because they fulfilled one or more exclusion criteria. Therefore, in the study, 68 patients were enrolled. Patients underwent a clinical assessment and their past medical history was recorded (i.e., evaluation of comorbidities, concomitant drug prescriptions, previous major surgeries, and the history of the disease that caused spasticity). At enrolment, all patients were assessed for: (a) MAS, which classifies the resistance opposed by the spastic muscle to passive mobilisation in six degrees (0 = normal tone, 4 = fixed retraction) [28] and (b) Modified Rankin Scale (MRS) a clinician-reported measure of global disability assessed using a seven-point scale (0 = no disability; 7 = high level of disability) [29]. Moreover, a patients’ perception-centred questionnaire was also administered in order to assess which part of the body caused greater discomfort, pain, and disability. The questionnaire explored three domains: (1) symptoms and impairment; (2) activities and function; (3) mobility/transfer and locomotion [30].

Based on the above, the selection of group of muscles (MGs) to be treated was done according to: class 1+ or greater from MAS score, the limb spasticity pattern, and patients’ needs.

Each participant underwent intramuscular infiltration of Botulinum Toxin Onabotulinum (Botox) or alternatively with Incobotulinum (Xeomin); the mean dose used was 450 IU for both drugs.

As recommended for Post Stroke Spasticity treatment [31], doses for individual muscles ranged from 10 to 100 U and total doses per posture was from 50 to 200 U for the upper limb [32], from 20 to 150 U for individual muscles, and 50 to 300 U for limb postures in the lower limb [33].

The patient lay in supine or pronated position, depending on the muscular group to be treated (Appendix A). The muscular groups treated were: 1. head and neck; 2. shoulder; 3. elbow extensors; 4. elbow flexors; 5. wrist and fingers flexors; 6. trunk; 7. hip flexors; 8. hip extensors; 9. hip adductors; 10. leg flexors; 11. foot plantar flexors; 12. foot dorsal flexors.

The infiltration was performed using a syringe with a 23-gauge, 30 mm needle, following a careful skin disinfection with a 70% alcohol swab. An ultrasound guide (Esaote Mylab Alpha ultrasound system) and 12–18 Mhz linear probe were used as a guide for the needle, which was positioned with a 45-angle degree to the skin surface and to the ultrasound probe.

### Statistical Analysis

Continuous variables were reported as mean and standard deviation (SD), whereas dichotomous variables were reported as number and percentages. Participants were divided into two groups according to sex and pattern of disability (classified as (a) hemiparesis, (b) spastic diplegia, and (c) double-hemiplegia).

Differences between the groups were evaluated by analysis of variance and chi-square test, for continuous and categorical variables, respectively. The chi-square test for linear trend was applied to estimate the probability that observed changes in an apparent trend representing true differences rather than chance findings.

To assess the association between sex or pattern of disability and BoNTA-treated muscles, independently from potential confounders, several different Logistic Regression Models were analysed, only for those muscles that in the univariate approach the *p*-value was less or equal to 0.10. Results were reported as odds ratio (OR) and 95% confidence intervals (95%CI) in Table 1 and Table 2. Confounders considered in Logistic models were: age, urinary incontinence, menstruation, and diagnosis. All statistical analyses were performed using SAS software rel.9.4.

## Figures and Tables

**Figure 1 toxins-14-00246-f001:**
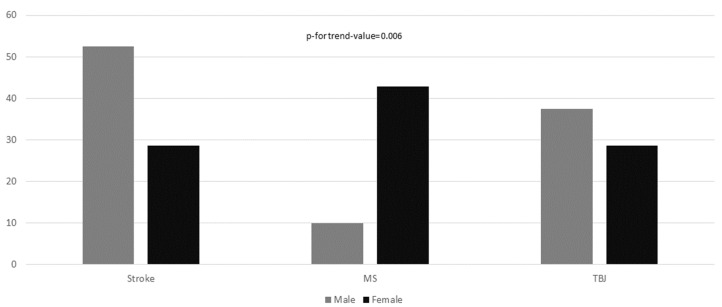
Disease prevalence according to sex. The chi-square test for linear trend was applied to estimate the probability that observed changes in an apparent trend representing true differences rather than chance findings.

**Figure 2 toxins-14-00246-f002:**
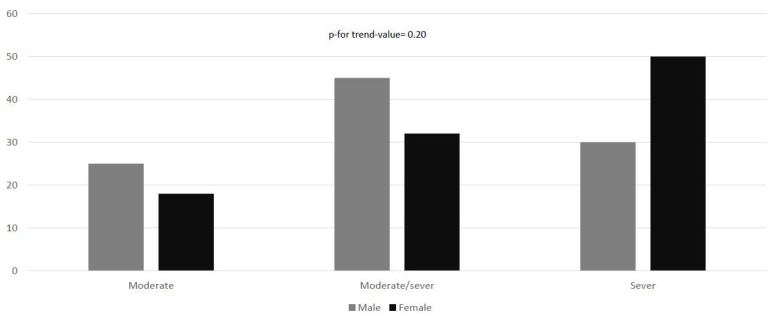
Modified Rankin disability scale according to sex. The chi-square test for linear trend was applied to estimate the probability that observed changes in an apparent trend representing true differences rather than chance findings.

**Table 1 toxins-14-00246-t001:** Differences in botulinum-treated muscles according to gender.

Variables	Male	Female		
	40	28	*p*-value	OR 95%CI *
Head	0 (0.0)	2 (7.1%)	0.17 ^	
Shoulder	16 (40.0)	10 (35.7)	0.72	
Triceps	7 (17.5)	1 (3.6)	0.08	0.28 (0.03–2.73)
Forearm	24 (60.0)	14 (50.0)	0.41	
Wrist	29 (72.5)	17 (50.0)	0.06	0.41 (0.15–1.15)
Trunk	1 (2.5)	4 (14.3)	0.07 ^	5.96 (0.62–57.40)
Hip	9 (22.5)	13 (46.4)	0.03	2.78 (0.94–8.26)
Gluteus	1 (2.5)	1 (3.6)	0.79 ^	
Adductor	5 (12.5)	15 (53.6)	<0.001	5.33 (1.42–20.09)
Crux	12 (30.0)	8 (28.6)	0.89	
Plantar	25 (62.5)	12 (42.9)	0.11	
Dorsal	1 (2.5)	1 (3.6)	0.79 ^	

* Odds Ratio and 95% Confidence Interval, adjusted for age, urine incontinence symptoms, menstruation cycle, and diagnosis. ^ χ^2^ Fischer exact test.

**Table 2 toxins-14-00246-t002:** Differences in botulinum-treated muscles according to expressivity of pattern disability. Spastic diplegia and double-hemiplegia versus Hemiparesis.

Variables	Spastic Diplegia and Double-Hemiplegia	Hemiparesis		
	27	41	*p*-value	OR 95%CI *
Head	0 (0.0)	2 (7.1)	0.36 ^	
Shoulder	4 (14.8)	22 (53.6)	0.002 ^	8.25 (1.88–36.18)
Triceps	2 (7.4)	6 (14.6)	0.47 ^	
Forearm	9 (33.3)	29 (70.7)	0.002	7.03 (1.90–25.97)
Wrist	10 (37.0)	33 (80.5)	<0.001	6.07 (1.72–21.34)
Trunk	5 (18.5)	0 (0.0)	0.008 ^	0.01 (0.01–999)
Hip	10 (37.4)	12 (29.3)	0.50	
Gluteus	1 (3.7)	1 (2.4)	0.99	
Adductor	16 (59.3)	4 (9.8)	<0.001 ^	0.12 (0.03–0.50)
Crux	12 (44.4)	8 (19.5)	0.03	0.13 (0.03–0.57)
Plantar	12 (44.4)	25 (61.0)	0.18	
Dorsal	1 (3.7)	1 (2.4)	0.99	
N° mm treated	3.0 ± 1.6	3.5 ± 1.4	0.18	

* Odds Ratio and 95% Confidence Interval, adjusted for age and sex. ^ χ^2^ Fischer exact test.

## Data Availability

The datasets used and/or analysed during the current study are available from the corresponding author on reasonable request.

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
