# Peer review of "An Observational Cross-Sectional Study of Gender and Disability as Determinants of Person-Centered Medicine in Botulinum Neurotoxin Treatment of Upper Motoneuron Syndrome"

_toxins, 2022, doi:10.3390/toxins14040246_

Round 1

Reviewer 1 Report

The concept of this paper is outstanding.  However, there are some key limitations, as described below:

  1. In the introduction, the authors say that "The treatment of adult spastici paresis now tends to focus on the treatment of muscle overactivity rather than...".  I believe that all of the treatment guidelines emphasize the importance of patient-centric treatment with a comprehensive approach to care.
  2. The methods in section 6.1 discuss a prespecified multivariate analysis and several different Logistic Regression Models.  These are not found in the Results section and they are key to understanding any conclusions. Baseline and change from baseline data is also missing.  In this situation, which is data-rich, a thorough analysis and presentation of the results is needed to justify the conclusions.

Author Response

  1. We appreciate the suggestion, and we agree with this reviewer consideration. Tailored medicine could not be considered as an option, and for this reason our study was centered on the unmet need of patients according to gender differences. To the best of our knowledge, in this area no guidelines clearly suggest how to approach at this problem.
  2. We have to thank the reviewer; we have made some changes in the description of the statistic section to better clarify where the results of the Logistic models were reported (table 1 and 2);
  3. Again, we have to thank the reviewer; to clarify the cross-sectional design of the study we change a little the title. As matter of fact the analysis presented were derived from an open outpatient service cohort, but in this paper our approach has a cross-sectional design.

Reviewer 2 Report

In the present paper, the authors evaluated which muscle groups were required to be treated with Botulinum neurotoxin A (BoNTA) injections to reduce spasticity in a cohort of patients. They found that the muscles that required Botulinum treatment were sex-gender-specific and were not only dependent on the clinical features of the disability.

The findings of this manuscript are useful because dysfunction of those muscles could influence the quality of life of the patients.

Minor text revisions are required before publication. For instance, the abbreviation used for Botulinum neurotoxin A should be the same in the whole manuscript (BoNTA, not BONTA like in Row 25, not BontA like in Row 149). The abbreviation “ASC” used in the Figure 1 should be mentioned in the Abbreviations Section.

Author Response

We thank the Reviewer for his/her kind comments. We apologies for those typos, and we have reanalyzed the paper to check for further incongruence

Reviewer 3 Report

The clinical intervention of BoNT in UMN syndrome is an area of research from very long time. BoNT is an excellent treatment options for muscle overactivity and spasticity. However, spasticity is the only one of the symptoms of UMN syndrome. Therefore, the treatment of BoNT may not be sufficient to have demonstrable effect on motor function. This paper didn't address this aspect of BoNT treatment. Although the author made an excellent analysis of sex related differences.

  1. Please use same abbreviation style.
  2. What is ASC in fig. 1? Figure should be more destructive.
  3. It would be better if we have the data of patients with different types of disease such as cerebral palsy, MS, SCI, stroke etc. and the effect of BoNT on UMN associated with these.
  4. Discussion could be more elaborative in terms of data created by the authors. They cite the other factors, but it is not clear whether they did  include these factors in their study.
  5. It is also not clear whether these effects are due to BoNT alone or associated with physical therapy.

Author Response

  1. We apologies for those typos, and we have reanalyzed the paper to check for further incongruence;
  2. Also in this case was a typo, and we have emended it;
  3. We appreciate the suggestion, but in our outpatient cohort the patients with a SCI were a very little sample, therefore we did not include them in this analysis. We reanalyzed data according to this reviewer suggestion, including in the Logistic models the disease that characterized UMNS as a confounding factor; results did not change substantially. Moreover, we agree with this Reviewer BoNTA-treatment efficacy was clearly and largely demonstrated, therefore our paper was focused in the evaluation of patients-need induced by spasticity according to gender.
  4. We did not understand the problem that was highlighted from this reviewer, we did not cite in the text “other factors”, as matter of fact in the text we use twice the word “factors”: 1) at page 6 taking into account the role of WASH, and we discussed the role of menses and incontinence; 2) page 6 again in the limitations, where we acknowledge and discussed the limitations of our study. We hypothesize that the problem could be in the description of the statistic approach and therefore we better specify in the statistical section which are the confounders considered in the logistic models.

We really appreciate the suggestion of this reviewer, as matter of fact we did not clearly state that all patients underwent physical therapy according to Rehab Individual Project that was drafted by the rehab-team. In the methods section we clarify this point (page 7). Therefore, the gender-effect we found in our analysis is totally ascribable to patients-needs, and is not mediated by physical rehab-procedure.

Reviewer 4 Report

Responses to Reviewer Comments

The reviewers are very grateful to the authors for their meticulous writing of the manuscript and the attractive ideas. The reviewers would like to kindly acknowledge the author’s work.

 As a medical doctor, and physical therapist, patients with upper motor neuron syndrome is well described as muscle hypertonia, or spasticity. The research conducted has significance that it is the first paper to assess the needs of patients that could benefit from botulinum neurotoxin treatment according to sex and to disability level. The patients with hemiparesis more frequently need to be treated in the upper limb; whereas, the diplegia/double-hemiparesis group needed to be treated more frequently at the adductor and crux muscles.

UMNS-spasticity in women could require more attention to be paid to the treatment of adductor muscle spasticity, potentially because the dysfunction of those muscles could influence sphincteric management, required for perineal hygiene and/or sexual life. The study have well-presented importance of sex based treatment and disability level of spasticity.

However, there needs to have figures that explains the treatment process of injection points, clinical pictures, for the readers of the “Toxins” are not familiar with the subject.

Again, thank you for the great research.

Author Response

We thank the Reviewer for his/her kind comments. We appreciate the suggestions of the referee, and we also include 3 supplementary figures and a short video of the process.

Round 2

Reviewer 1 Report

Comments have been addressed.  Please provide a little more context for Figures 1 and 2 regarding the "p for trend".  It is commonly used in epidemiologic studies, but the readership of Toxins may not be as familiar.  This could be briefly noted either in the Stats section or the discussion.  Probably needs just 1-2 sentences.

Author Response

We appreciate this reviewer sujestion, and according to it we insert a sentence both in method-section as in the footnotes of the figure. For a consistency in the paper we use the same definition.

Thanks